# Dynamics of ranking

Gerardo Iñiguez [1,2,3✉], Carlos Pineda [4], Carlos Gershenson [3,5,6,7] & Albert-László Barabási[1,7,8✉]

Virtually anything can be and is ranked; people, institutions, countries, words, genes. Rankings reduce complex systems to ordered lists, reflecting the ability of their elements to perform relevant functions, and are being used from socioeconomic policy to knowledge extraction. A century of research has found regularities when temporal rank data is aggregated. Far less is known, however, about how rankings change in time. Here we explore the dynamics of 30 rankings in natural, social, economic, and infrastructural systems, comprising millions of elements and timescales from minutes to centuries. We find that the flux of new elements determines the stability of a ranking: for high flux only the top of the list is stable, otherwise top and bottom are equally stable. We show that two basic mechanisms — displacement and replacement of elements — capture empirical ranking dynamics. The model uncovers two regimes of behavior; fast and large rank changes, or slow diffusion. Our results indicate that the balance between robustness and adaptability in ranked systems might be governed by simple random processes irrespective of system details.

---

[1] Department of Network and Data Science, Central European University, 1100 Vienna, Austria. [2] Department of Computer Science, Aalto University School of Science, 00076 Aalto, Finland. [3] Centro de Ciencias de la Complejidad, Universidad Nacional Autonóma de México, 04510 Ciudad de México, Mexico. [4] Instituto de Física, Universidad Nacional Autonóma de México, 04510 Ciudad de México, Mexico. [5] Instituto de Investigaciones en Matemáticas Aplicadas y en Sistemas, Universidad Nacional Autonóma de México, 04510 Ciudad de México, Mexico. [6] Lakeside Labs GmbH, Lakeside Park B04, 9020 Klagenfurt am Wörthersee, Austria. [7] Network Science Institute, Center for Complex Network Research & Department of Physics, Northeastern University, 02115 Boston, MA, USA. [8] Channing Division of Network Medicine & Department of Medicine, Brigham and Women's Hospital, Harvard Medical School, 02215 Boston, MA, USA. ✉email: iniguezg@ceu.edu; alb@neu.edu

Rankings are everywhere. From country development indices, academic indicators, and candidate poll numbers to music charts and sports scoreboards, rankings are key to how humans measure and make sense of the world[1,2]. The ubiquity of rankings stems from the generality of their definition: reducing the (often high-dimensional) complexity of a system to a few or even a single measurable quantity of interest[3,4], dubbed score, leads to an ordered list where elements are ranked, typically from highest to lowest score. Rankings are, in this sense, a proxy of relevance or fitness to perform a function in the system. Rankings are being used to identify the most accomplished individuals or institutions, and to find the essential pieces of knowledge or infrastructure in society[1]. Since rankings often determine who gets access to resources (education, jobs, and funds), they play a role in the formation of social hierarchies[5,6] and the potential rise of systematic inequality[7].

The statistical properties of ranking lists have caught the attention of natural and social scientists for more than a century. A heavy-tailed decay of score with rank, commonly known as Zipf's law[8,9], has been systematically observed in the ranking of cities by population[10,11], words and phrases by frequency of use[12–17], companies by size[18–20], and many features of the Internet[21]. Zipf's law appears even in the score-rank distributions of natural systems, such as earthquakes[22,23], DNA sequences[24], and metabolic networks[25]. Rankings have also proven useful when analyzing productivity and impact in science and the arts[7,26–29], in human urban mobility[30–32], epidemic spreading by influentials[33], and the development pathways of entire countries[34]. Recently, studies of language use[17,35], sports performance[36], and many biological and socioeconomic rankings[37] have strengthened the notion of universality suggested by Zipf's law: despite microscopic differences in elements, scores, and types of interaction, the aggregate, macroscopic properties of ranking lists are remarkably similar throughout nature and society.

The similarity of score-rank distributions across systems raises the question of the existence of simple generative mechanisms behind them. While mechanisms of proportional growth[38], cumulative advantage[39], and preferential attachment[40] are often used to explain the heavy-tailed distributions of ranking lists at single points of time[41,42], they fail to reproduce the way elements actually move in rank[43], such as the sudden changes in city size throughout history[44,45]. Here, we report on the existence of generic features of rank dynamics over a wide array of systems, from individuals to countries, and spatio-temporal scales, from minutes to centuries. By measuring the flux of elements across ranking lists[46–48], we identify a continuum ranging from systems where highly ranked elements are more stable than the rest, to systems where the least relevant elements are also stable. We show that simple mechanisms relying on fluxes generated by displacement and replacement of elements can account for all observed patterns of rank stability. A model based on these ingredients uncovers two regimes in rank dynamics, a fast regime driven by long jumps in rank space, and a slow one driven by diffusion.

## Results

We gather 30 ranking lists in natural, social, economic, and infrastructural systems. Data include human and animal groups, languages, countries and cities, universities, companies, transportation systems, online platforms, and sports, with no selection criteria other than having enough information for analysis (for data details see Supplementary Information [SI] Section S2 and Table S1). Elements in each list are ranked by a measurable score that changes in time: scientists by citations, businesses by revenue, regions by a number of earthquakes, players by points, etc.

Size and temporal scales in the data vary widely, from the number of people in 636 station entrances of the London Underground every 15min during a week in 2012[49], to the written frequency of 124k English words every year since the 17th century[50]. Following an element's rank through time reveals systematic patterns (Fig. 1). For example, in the Academic Ranking of World Universities (ARWU)[51], published yearly since 2003, institutions like Harvard and Stanford maintain a high score, while institutions down the list change rank frequently (Fig. 1a).

Ranking lists typically have a fixed size $N_0$ (e.g., the Top 100 universities[51], the Fortune 500 companies[52]), so elements may enter or leave the list at any of the $T$ observations $t = 0, ..., T - 1$, allowing us to measure the flux of elements across rank boundaries[42,46,47] (for the observed values of $N_0$, $T$ see SI Table S1). We introduce two time-dependent measures of flux: the *rank turnover* $o_t = N_t/N_0$, representing the number $N_t$ of elements ever seen in the ranking list until time $t$ relative to the list size $N_0$, and the *rank flux* $F_t$, representing the probability that an element enters or leaves the ranking list at time $t$. Rank turnover is a monotonic increasing function indicating how fast new elements reach the list (Fig. 1b left; all datasets in SI Fig. S5). In turn, flux shows striking stationarity in time despite differences in temporal scales and potential shocks to the system (SI Fig. S3). By averaging over time, the mean turnover rate $\dot{o} = (o_{T-1} - o_0)/(T - 1)$ and the mean flux $F = \langle F_t \rangle$ turn out to be highly correlated quantities that uncover a continuum of ranking lists (Fig. 1b right; values in SI Table S2). In one extreme, the most open systems ($F, \dot{o} \sim 1$) have elements that constantly enter and leave the list. Less open systems ($F, \dot{o} \sim 0$) have a progressively lower turnout of constituents. Five out of 30 ranking lists are completely closed ($F = \dot{o} = 0$), meaning no single new element is recorded during the observation window.

The measures of rank turnover and flux reveal regularities in the stability of ranking processes[43,53]. We follow the time series of the rank $R_t$ occupied by a given element at time $t$[44] (Fig. 1c; all datasets in SI Fig. S2). In most systems, highly ranked elements like Harvard University and the English word 'the' never change position, showcasing the correspondence between rank stability and notions of relevance like academic prestige[7,27], grammatical function[17,50], and underlying network structure[53]. As we go down the ranking list of open systems, rank trajectories increasingly fluctuate in time. In the least open systems where turnover and flux are low, however, low ranked elements are also stable. In the ranking of British cities by population, for example, both Birmingham and Nairn remain the most and least populated local authority areas throughout the 20th century[54]. These findings uncover a more fine-grained sense of rank stability: most systems have a stable top ranking, but only the least open systems feature stable bottom ranks as well. The rank change $C$, measured as the average probability that element at rank $R$ changes between times $t - 1$ and $t$, varies between an approximately monotonic increasing function of $R$ for open systems to a symmetric shape as systems become less open (Fig. 1d; all datasets in SI Fig. S6).

Since the stability patterns of an empirical ranking list (as measured by rank change $C$) can be systematically connected to the number of elements flowing into and out of the list, we build a model of rank dynamics based solely on simple generative mechanisms of flux (Fig. 2). Without assuming system-specific features of elements or their interactions, there are at least two ways to implement flux in rank space. Smooth (but arbitrarily large) changes in the score of an element might make it larger or smaller than other scores, causing elements to move across ranks (the way some scientists gather more citations than others[27], or how population size fluctuates due to historical events[44]). Regardless of the score, elements might also disappear from the list and be replaced by new elements: young athletes enter competitions while old ones retire[36]; new words replace

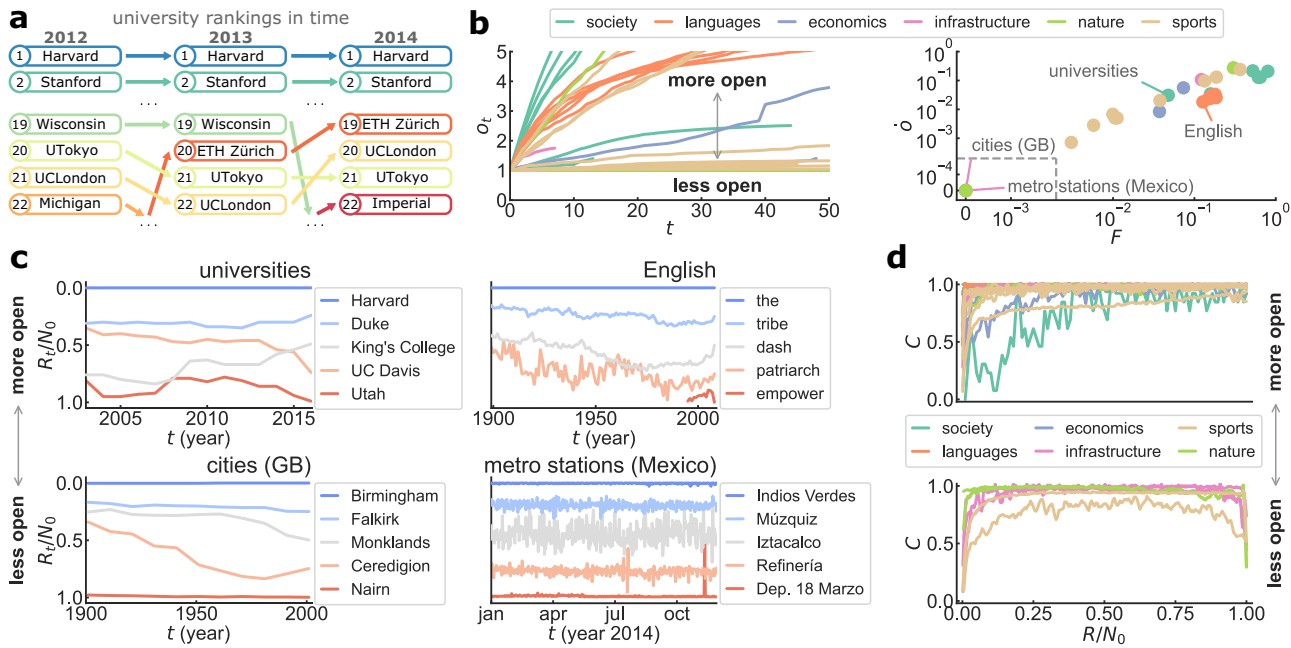

**Fig. 1 Ranking lists in nature and society show generic patterns in their dynamics. a** Yearly top ranking of universities worldwide according to ARWU score[51]. Elements in the system change rank as their scores evolve in time. **b** (Left) Rank turnover $o_t$ at time $t$ for studied systems, defined as the number $N_t$ of elements ever seen in the ranking list up to $t$ relative to list size $N_0$ (see SI Fig. S5). (Right) Correlation between mean turnover rate $\dot{o}$ and mean flux $F$ (average probability that an element enters or leaves the list). Ranking lists form a continuum from the most open systems ($F, \dot{o} \sim 1$) to the less open ($F, \dot{o} \sim 0$; for values see SI Table S2). The area between dashed lines has linear scales to show closed systems with $F = \dot{o} = 0$. **(c)** Time series of rank $R_t/N_0$ occupied by elements across the ranking list in selected systems (all datasets in SI Fig. S2). In the least open systems available, the top and bottom of the ranking list are stable. In open systems, only the top is stable. **d** Rank change $C$ (average probability that element at rank $R$ changes between $t-1$ and $t$) across ranking lists (see SI Fig. S6), for $F \geq 0.01$ (top) and $F < 0.01$ (bottom). The stable top and bottom ranks of less open systems mean $C$ is roughly symmetric. In open systems, $C$ increases with rank $R$.

anachronisms due to cultural shifts[50]. We implement random mechanisms of displacement and replacement in a simple model by considering a synthetic ranking list of length $N_0$ embedded within a larger system of size $N \geq N_0$. At each time step of length $\Delta t = 1/N$, a randomly chosen element moves to a randomly selected rank with probability $\tau$, displacing others. At the same time, a randomly chosen element gets replaced by a new one with probability $\nu$, leaving other ranks untouched. The dynamics involve all $N$ elements, but to mimic real ranking lists, we only consider the top $N_0$ ranks when comparing with empirical data (Fig. 2a; model details in SI Section S4).

We solve the model analytically by introducing the displacement probability $P_{x,t}$ that an element with rank $r = R/N$ gets displaced to rank $x = X/N$ after a time $t$ (Fig. 2b; uppercase/lowercase symbols denote integer/normalized ranks). Since for small $\Delta x = 1/N$ the probability that at time $t$ an element has not yet been replaced is $e^{-\nu t}$, we have

$$P_{x,t} = e^{-\nu t}(L_t + D_{x,t}). \quad (1)$$

Here, $L_t = (1 - e^{-\tau t})/N$ is the (rank-independent) probability that up until time $t$ an element gets selected and jumps to any other rank. The length of jumps is uniformly distributed, so they can be thought of as a Lévy random walk with step length exponent 0[55] (full derivation in SI Section S4). The probability $D_{x,t} = D(x, t)\Delta x$ that the element in rank $r$ gets displaced to rank $x$ after a time $t$ (due to Lévy walks of other elements) follows approximately the diffusion-like equation

$$\frac{\partial D}{\partial t} = \alpha x(1 - x)\frac{\partial^2 D}{\partial x^2}, \quad (2)$$

where $\alpha = \tau/N$. Since $\sum_x D_{x,t} = e^{-\tau t}$, both $D_{x,t}$ and $D(x, t)$ are not conserved in time. Instead of a standard diffusion equation, Eq.

(2) is equivalent to the Wright-Fisher equation of random genetic drift in allele populations[56,57]. The solution $D(x, t)$ of Eq. (2) is well approximated by a decaying Gaussian distribution with mean $r$ and standard deviation $\sqrt{2\alpha r(1 - r)t}$, i.e., a diffusion kernel (Fig. 2b). Overall, local displacement makes elements slowly diffuse around their initial rank, while Lévy walks and the replacement dynamics reduces exponentially the probability that old elements remain in the ranking list.

An explicit expression for the displacement probability $P_{x,t}$ allows us to derive the mean flux

$$F = 1 - e^{-\nu}[p + (1 - p)e^{-\tau}], \quad (3)$$

and the mean turnover rate

$$\dot{o} = \nu\frac{\nu + \tau}{\nu + p\tau}, \quad (4)$$

where $p = N_0/N$ is the length of the ranking list relative to system size (see SI Section S4). In order to fit the model to each empirical ranking list, we obtain $N_0$ from the data and approximate $N = N_{T-1}$ as the number of distinct elements ever seen in the list during the observation period $T$, thus fixing $p$ (values for all datasets in SI Tables S1–S2). The remaining free parameters $\tau$ and $\nu$ (regulating the mechanisms of displacement and replacement) come from numerically solving Eqs. (3)–(4) with $F$ and $\dot{o}$ fixed by the data (Figs. 1b and 2c; for model fitting see SI Section S5). The approximations in Eqs. (3)–(4) introduce a small bias in the estimation of $\tau$ (SI Fig. S18). Despite this bias, the simple generative mechanisms of flux in the model are enough to recover the behavior of ranking lists as quantified by $P_{x,t}$ and $C$ (Fig. 2d and SI Fig. S19): When rank flux is low, both the top and bottom of the list are similarly stable and rank dynamics is mostly driven by an interplay between Lévy walks and diffusion. As systems become more open, however, this

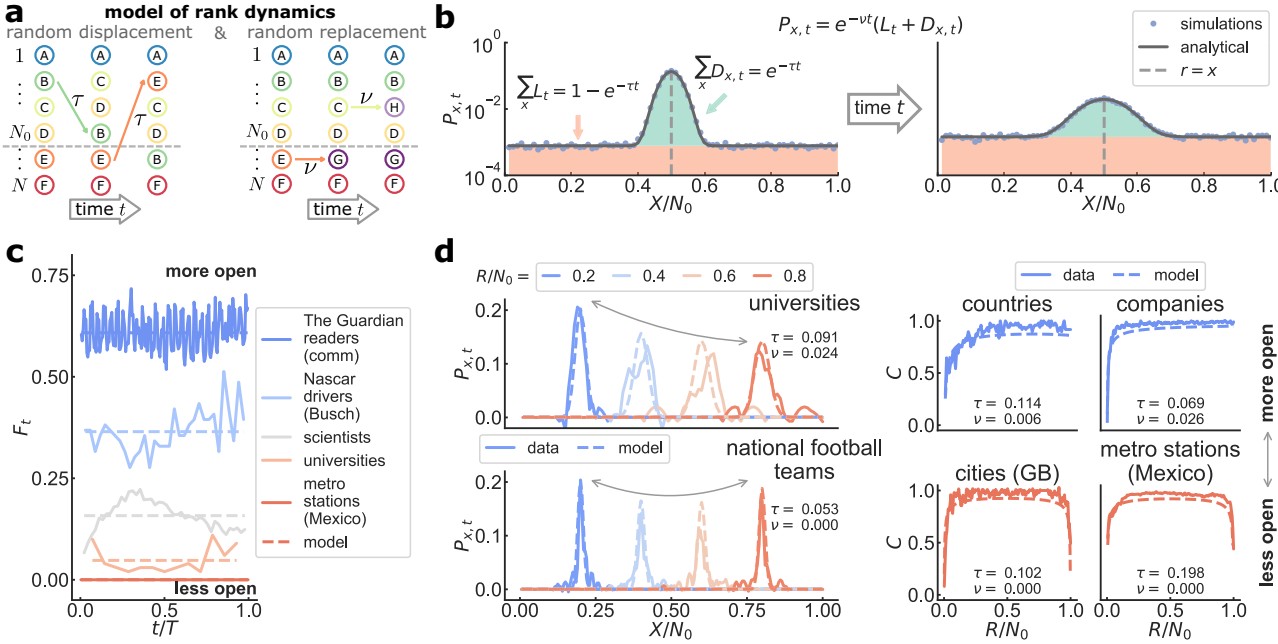

**Fig. 2 Model of rank dynamics reproduces features of real-world ranking lists. a** Model of rank dynamics in a system of $N$ elements and ranking list size $N_0$. At time $t$, a random element is moved to a random rank with probability $\tau$. A random element is also replaced by a new element with probability $\nu$. **b** Probability $P_{x,t}$ that element in rank $r = R/N$ moves to $x = X/N$ after time $t$ (uppercase/lowercase symbols are integer/normalized ranks). Elements not replaced diffuse around $x = r$ (with probability $D_{x,t}$) or perform Lévy walks[55] (with probability $L_t$). Eq. (2) recovers simulation results, shown here for $\tau = 0.1$, $\nu = 0.2$, $N = 100$, and $N_0 = 80$ at times $t = 1, 5$ (left/right plots), averaged over $10^5$ realizations. **c** Time series of rank flux $F_t$ over observation period $T$ for data (lines), and mean flux $F$ from the fitted model (dashes) (all datasets in SI Fig. S3; for fitting see SI Section S5). **d** Probability $P_{x,t}$ for $t = 1$ and varying $r$ (left) and rank change $C$ (right), shown for selected datasets (lines) and fitted model (dashes; $\tau$ and $\nu$ in the plot) (empirical $P_{x,t}$ is passed through a Savitzky–Golay smoothing filter; see SI Figs. S6–S9 and SI Table S2). As systems become more open, we lose symmetry in the rank dependence of both $C$ and the height of the diffusion peaks of $P_{x,t}$ (signaled by curved arrows). Data and model have similar qualitative behavior in all rank measures (for a systematic comparison see SI Fig. S19).

symmetry gets broken due to a growing flux of elements at the bottom of the ranking list (see SI Fig. S4). Regardless of whether we rank people or animals, words or countries, the pattern of stability across a ranking list is accurately emulated by random mechanisms of flux that disregard the microscopic details of the individual system.

The characterization of flux in ranking lists with mechanisms of displacement and replacement of elements reveals regimes of dynamical behavior that are not apparent from the data alone (Fig. 3). By rescaling the fitted parameter values of the model as

$$\tau_r = \frac{\tau}{p(1-p)\dot{o}} \quad \text{and} \quad \nu_r = \frac{\nu - p\dot{o}}{\dot{o}}, \tag{5}$$

most open ranking lists $(F, \dot{o} > 0)$ are predicted to follow the universal curve

$$\tau_r \nu_r = 1, \tag{6}$$

which suggests that ranking dynamics are regulated by a single effective parameter [Fig. 3a; derivation in SI Section S5; for a discussion of the role of fluctuations on the validity of Eq. (6) see SI Figs. S18–S20]. Even if, potentially, displacement and replacement could appear in any relative quantity, adjusting the model to observations of rank flux and turnover (Fig. 1b) leads to an inverse relationship between parameters regulating their generative mechanisms. Real-world ranking lists lie in a spectrum where their dynamics is either mainly driven by score changes that displace elements in rank (high $\tau_r$ and low $\nu_r$, like for GitHub software repositories[58] ranked by daily popularity), or by birth-death processes triggering element replacement (low $\tau_r$ and high $\nu_r$, like for Fortune 500 companies[52] ranked by yearly revenue). While the symmetry (or lack thereof) in rank change $C$ may

seemingly imply two distinct classes of systems (see Figs. 1d and 2d), Eq. (6) reveals the existence of a continuum of open ranking lists, which can be captured by a single model with a single effective parameter.

Data on empirical ranking lists is constrained by the average time length $\ell$ between recorded observations, which varies from minutes to years depending on the source and intended use of the rankings ($\ell$ for all datasets is listed in SI Table S1). We explore such scoping effect by subsampling data every $k$ observations (for details see SI Section S5 and SI Figs. S21–S22). Longer times between snapshots of the ranking list lead to an increase in rank flux, turnover, and fitted parameters, such that the rate of element replacement $\nu/k\ell$ stays roughly constant (Fig. 3b top). A conserved replacement probability per unit time, robust to changes in sampling rate, is yet another measure of rank stability: online social systems exchange elements frequently (e.g., the ranking lists by daily popularity of both GitHub software repositories and of online readers of the British newspaper The Guardian[59]), followed by sports, while languages are the most stable (values for $k = 1$ in SI Table S2).

The universal curve in Eq. (6) displays three regimes in the dynamics of open ranking lists, as measured by the average probabilities that, between consecutive observations in the data, an element performs either a Lévy walk $[W_{levy} = e^{-\nu}(1 - e^{-\tau})]$, changes rank by diffusion $[W_{diff} = e^{-\nu}e^{-\tau}]$, or is replaced $[W_{repl} = 1 - e^{-\nu}]$, with $W_{levy} + W_{diff} + W_{repl} = 1$. In systems with the largest rank flux and turnover (GitHub repositories and The Guardian readers), elements tend to change rank via long jumps, following a Lévy walk, where $W_{levy} > W_{diff}, W_{repl}$ (Fig. 3c). Here, long-range rank changes take elements in and out of a short ranking list within a big system (low $p$), thus generating large

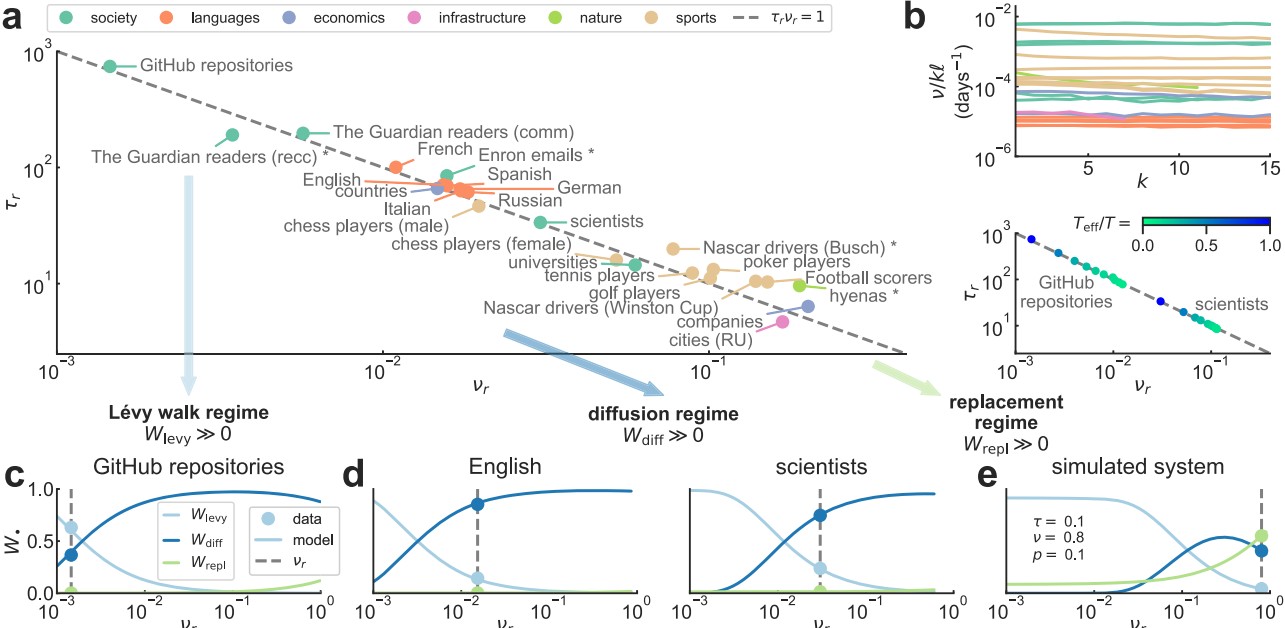

**Fig. 3 Model uncovers regimes of dynamical behavior in open ranking lists. a** Rescaled model parameters $\tau_r$ and $\nu_r$ in open ranking lists, obtained from fitted parameters $\tau$ and $\nu$, relative ranking list size $p$, and mean turnover rate $\dot{o}$ [see Eq. (5) and SI Section S5; only systems with $\dot{o} > 10^{-3}$ are shown]. Values collapse onto the universal curve $\tau_r \nu_r = 1$, so an inverse relationship between displacement and replacement is enough to emulate empirical rank dynamics (asterisks denote datasets that are farther away from the universal curve than bootstrapped model simulations; see SI Fig. S20). **b** (Top) Rate of element replacement $\nu/k\ell$ when subsampling data every $k$ observations of length $\ell$ (see SI Table S1 and SI Section S5). Online social systems have the largest rates, followed by sports and languages (SI Table S2). (Bottom) Parameters $\tau_r$ and $\nu_r$ for $T_{eff} = \lceil T/k \rceil$ subsampled observations (all datasets in SI Fig. S22). By subsampling ranking dynamics, systems move downwards along the universal curve while keeping a constant replacement rate. **c–e** Average probability that an element changes rank by Lévy walk ($W_{levy}$), diffusion ($W_{diff}$), or is replaced ($W_{repl}$) between consecutive observations in the data. Probabilities are shown both for selected datasets (dots), and for the model moving along the curve $\tau_r \nu_r = 1$ with the same $p$ and $\dot{o}$ as the data (lines) (for rest of systems see SI Fig. S17). The simulated system in (**e**) is the model itself for given values of $\tau$, $\nu$, and $p$ (shown in plot). The model reveals a crossover in real-world ranking lists between a regime dominated by Lévy walks (**b**) to one driven by diffusion (**c**). Although not seen in data, the model also predicts a third regime driven by replacement (**d**).

mean flux $F$ (see SI Table S2). Most datasets, like the yearly rankings of scientists by citations in American Physical Society journals[27,60] and of countries by economic complexity[34,61], belong instead to a diffusion regime with $W_{diff} > W_{levy}, W_{repl}$ (Fig. 3d). In this regime, local, diffusive rank dynamics is the result of elements smoothly changing their scores and overcoming their neighbors in rank space. Under subsampling, ranking lists move downwards along the universal curve, going from a state with a certain number of Lévy walks to one more driven by diffusion (Fig. 3b bottom; all datasets in SI Fig. S22).

The model also predicts a third regime dominated by replacement ($W_{repl} > W_{levy}, W_{diff}$; Fig. 3e), where elements are more likely to disappear than change rank. Such ranking lists replace most constituents from one observation to the next, forming a highly fluctuating regime that we do not observe in empirical data. To showcase the crossover between regimes, we simulate the model along the universal curve of Eq. (6) while keeping $p$ and $\dot{o}$ fixed in Eq. (5) (lines in Fig. 3c–e). These curves show how close systems are to a change of regime, i.e., from one dominated by Lévy walks to one driven by diffusion. When a ranking list is close to a regime boundary, external shocks (amounting to variations in parameters $\tau$ and $\nu$) may change the main mechanism behind rank dynamics, thus affecting the overall stability of the system.

## Discussion

Ranking lists reduce the elements of high-dimensional complex systems into ordered values of a summary statistic, allowing us to compare seemingly disparate phenomena in nature and society[2,42]. The diversity of their components (people, animals, words, institutions, and countries) stands in contrast with the statistical regularity of score-rank distributions when aggregated over time[9,37]. By exploring the flux of elements of 30 ranking lists in natural, social, economic, and infrastructural systems, we present evidence of generic temporal patterns of rank dynamics. While open systems (large flux) keep the same elements only in top ranks, less open systems (lower flux) also have stable bottom ranks, forming a continuum of ranking lists explained by a single class of models. The model reveals two regimes of dynamical behavior for systems with nonzero flux. Real-world ranking lists are driven either by Lévy random walks[55] that change the rank of elements abruptly or by a more local, diffusive movement similar to genetic drift[56,57], both alongside a relatively low rate of element replacement independent from the frequency at which the ranking list is measured.

Our results suggest that, even though score distributions differ across systems depending on what type of elements and interactions they have (SI Fig. S1), ranking lists have similar stability features. What are the underlying properties of the system that enhance this similarity? An extension of our model explicitly considering the links between score and rank may help further understand the experimental evidence in this area, like the recent observation that the stability of crowdsourced rankings depends on the magnitude difference between quality scores[62]. It is also interesting to consider the observed deviations from the predictions of our model, even at the level of ranks. Rank flux for languages is not constant but decreases over time (SI Fig. S3), arguably due to the long observation period (over three centuries;

see SI Table S1), variations in word sampling across decades, or even cognitive distortions at the societal scale[63]. The rank-dependence of flux for very open systems (SI Fig. S4) and the slow decay of inertia with long times we observe in most datasets (SI Fig. S7) might be better reproduced by a non-uniform sampling of elements in the mechanisms of displacement and replacement of the model. Finally, deviations in the data (indicating a departure from the assumptions of randomness and stationarity built into our model) could be used to detect shocks to the system larger than expected statistical fluctuations, such as the sudden increase in the rank flux of Fortune 500 companies during financial crises (SI Fig. S3).

A more nuanced understanding of the generic features of ranking dynamics might help us limit resource exhaustion in competitive environments, such as information overload in online social platforms and prestige biases in scientific publishing[64], via better algorithmic rating tools[65]. The observation of a systematic interplay between "slower" and "faster" ranking dynamics[66] (see SI Section S6) can be refined by exploring the relationship between ranking lists associated with the same system, or by incorporating networked interactions that lead to macroscopic ordering[44,67], which may provide a deeper understanding of network centrality measures based on ranking[68]. Given that rankings often mediate access to resources via policy, similar mechanisms to those explored here may play a role in finding better ways to avoid social and economic disparity. In general, a better understanding of rank dynamics is promising for regulating systems by adjusting their temporal heterogeneity.

## Data availability
For data availability, see SI Section S2. Non-public data is available from the authors upon reasonable request.

## Code availability
Code to reproduce the results of the paper is publicly available at https://github.com/iniguezg/Farranks[69].

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

## Acknowledgements

In memory of Jorge Flores and Germinal Cocho. We acknowledge José A. Morales and Sergio Sánchez for data handling at the start of the project. We are grateful for data provision to Gustavo Carreón, Syed Haque, Kay Holekamp, Amiyaal Ilany, Márton Karsai, Raj Kumar Pan, Roberto Murcio, and Roberta Sinatra. G.I. thanks Tiina Näsi for valuable suggestions. G.I. acknowledges support from AFOSR (#FA8655-20-1-7020), project EU H2020 Humane AI-net (#952026), and CHIST-ERA project SAI (#FWF I 5205-N). C.P. and C.G. acknowledge support by CONACyT (#285754) and UNAM-PAPIIT (#IG100518, IG101421, IN107919, and IV100120). C.G. was also supported by the PASPA program from UNAM-DGAPA. A.-L.B. was supported by an EU H2020 SYNERGY grant (#810115-DYNASNET), the John Templeton Foundation (#61066), and AFOSR (#FA9550-19-1-0354).

## Author contributions

G.I., C.P., C.G., and A.-L.B. designed the study. G.I. performed data analysis and model fitting. G.I. and C.P. derived analytical results and performed numerical simulations. G.I., C.P., C.G., and A.-L.B. wrote the paper.

## Competing interests

A.-L.B. is the founder of Foodome, ScipherMedicine, and Datapolis, companies that explore the role of networks in health and urban environments. G.I. is the founder of Predify, a data science consulting startup in Mexico. C.P. and C.G. declare no competing interest.
