## [Peer Review File · Nature Communications]

REVIEWER COMMENTS

Reviewer #1 (Remarks to the Author):

First of all, I want to thank the authors for a paper that I enjoyed reading. It investigates the dynamic of ranking in a large set of empirical data. It proposes an attractive model based on the notion of evolution or revolution. The evolution process is a controlled process of diffusion. The revolution process consists of long ranking jumps. Results of the experimentation show that real-world systems can be characterized by their rank openness and rank flux. Rank openness measures how fast new elements enter the ranking list. Rank flux estimates the probability that an element enters or leaves the ranking list at a given time. No new elements enter the list in closed systems, while they constantly enter and exit in open systems. The authors show that the top ranks are conserved in real-world open systems. Elements ranked at the top and the bottom of the list is stable in closed systems. They also show that displacement and replacement of elements are sufficient to control this process. The paper is well written and well-organized. The empirical study is very rich. The results are very clearly presented. It will indeed generate interesting discussions on centrality concepts and beyond. These are the strong points for publication. My minor concerns are the following:

- 1) I am always a little skeptical about universal laws. I understand that the authors want to put under the same umbrella phenomena of diverse origin. I think it will be interesting to comment a bit on the eventual deviations that may have appeared in the experiments.
- 2) The authors claim, "Since rankings mediate access to resources (education, jobs, funds) via public and economic policy, they play a role in social mobility; when misused, they contribute to the disenfranchisement of minorities." I did not fully understand this remark. I think that they should give more explanations or remove this sentence.
- 3) Let's consider that there are multiple jumps at various ranges combined with multiple diffusion at multiple scales. How will that affect the model?
- 4) Not all readers are familiar with the Savitzky-Golay filter, so it is worth noticing that it is a smoothing filter.

Reviewer #2 (Remarks to the Author):

This manuscript argues that a simple model can recreate simple patterns in the dynamics of rankings. It is an excellent piece of research—at the same time accessible and profound. I recommend it for publication in Nature Communication after some minor changes.

[Openness rate] I think it would be much more intuitive to call this "turnover rate," and unless the authors have a perfect excuse, please change this. Essentially, this is just the colloquial understanding of turnover rate, whereas openness could mean that the ranking rules are open to change or whatever.

[Citations] This paper seems relevant. It is published after the authors submitted the present manuscript, so I don't criticize its omission, but for the reader, it would be helpful if the authors could discuss it:

Mari Kawakatsu, Philip S. Chodrow, Nicole Eikmeier, Daniel B. Larremore, "Emergence of hierarchy in networked endorsement dynamics"
Proceedings of the National Academy of Sciences Apr 2021, 118 (16) e2015188118

[When considering open systems in close interaction with their surroundings...] I think in this paragraph, the authors get carried away too far beyond what their results suggest. "[S]lower' elements provide robustness, while 'faster' elements provide adaptivity." ... for this to connect to grander themes of biology and evolution, the ranking in its entirety must

have some meaning, like a genetic code or similar. However, in practice, unlike such a code, only the top of a ranking matters. I think the results in this work are great without speculations like this, so the paper would be stronger if the authors deleted the paragraph.

Reviewer #3 (Remarks to the Author):

Summary

This is a review of "Universal dynamics of ranking" submitted to Nature Communications for consideration. In this paper, the authors consider two simple mechanisms which describe a dynamics over ordered lists. Either (1) an item from one position is moved to an arbitrary position in the list, chosen uniformly at random, with items between the two positions shifted to account for the move, or (2) an item is deleted and replaced with a new item. The first occurs with probability τ and the second with probability ν , in each time step; the moves are not mutually exclusive. Finally, the authors assume that we do not see ALL of the list (N elements) but only the first N_0 . Through a series of asymptotic approximations, the authors arrive at Eqs (5) and (6) which argue for a universal inverse relationship between τ and ν , after each has been rescaled by p (the relative length of the visible list, N_0/N) and o (the average number of new elements observed over the period of observation). When the authors examine 30 real datasets, fit all their parameters, and plot them in rescaled $\log(\tau)$ $\log(\nu)$ space, the points lie around the line. This leads to the conclusion that there exists a universal dynamics for ranking.

Through the notes that follow, I cannot recommend acceptance of the paper, but hope that the comments are nevertheless useful.

Major Comments

After reading the paper and supplement, I'm not sure what it actually means for understanding real world systems or the dynamics of sorted lists. The top-line conclusion is that real systems must have either high replacement and low movement, or high movement and low replacement, or some combination along the $\text{replacement} \times \text{movement} = 1$ line... but after rescaling by the mean flux, the observed and unobserved system sizes. It sounds like the conclusion is that *any* set of ranking observations, generated by any process or model would, after embedding according to this model, fall along the $\text{nur} \times \text{taur} = 1$ line, more or less. So does the more or less matter? Do we learn something from the fact that Nascar Drivers and Hyenas lie far from the universal line while the languages lie near it? Are these interesting fluctuations or expected fluctuations? The numerous plots of the supplement indeed show that approximations can recapitulate the means of simulations, but are the fluctuations around the means large or meaningful? In short, this manuscript presents a model as a way to gain understanding of the dynamics of rankings, but the analysis and embedding of the model don't actually end up providing that understanding. In this light, it is hard to see how the paper will be of interest to or influence the field.

The conclusion that "Real world ranking lists are drive either by [large jumps], or [small local movements], both alongside relatively low levels of new elements entering the system," is true, with or without the study. The last clause, regarding the rate at which new elements enter, seems entirely dependent on the frequency of sampling. The remaining paragraphs of the discussion read as broadly speculative. In short, the claims of the paper seem really bold but thinly supported.

More constructively, I suggest that the authors consider a venue that allows their analysis more room to breathe. The paper was not understandable without the supplementary

material, but the supplementary material was clearly written and really explicated the model nicely. Sections S3 and S4 were engaging and interesting. I very much enjoyed reading the mathematics of the paper, yet the density of the writing and notation really got in the way of the communication of ideas, leading me to create notecards to follow along. For instance, what is called a Levy or Levi flight with power law exponent 0 is a complicated way of saying that an element moves to a position in the list chosen uniformly at random.

Another suggestion to improve the analysis would be to consider whether there are available statistical approaches to model fitting, rather than graphical/numerical fits to asymptotic approximations of mean behavior. Given a dataset, and the authors' posed generative process, could one simply find MLE estimates of tau and nu? If the generative process is available, could one determine whether real systems are within the expected fluctuations of the model or whether the dynamics doesn't apply well?

Finally, the authors have the ability to evaluate how real data points move along the "universal" curve in the presence of subsampling (say, discarding every other or every third sample of ranks) or arbitrarily shifting upward the line at which rank flux is calculated. Showing how the inferred tau and nu shift, for a single underlying system, when viewed through different censoring or scoping, would help future readers understand how sampling rate and sample scope of the data affects the inferences one draws from it.

Minor comments

- As mentioned above, I suggest the authors consider not invoking Levy flights. If they feel it is necessary to do so, I suggest consistently writing Levi or Levy, but not both.
- S4.1 the idea of "left/right" is introduced before it is explained in S4.2. A simple reordering could fix.
- Though identical, perhaps an simpler alternative to deriving S8 is to go directly at it: the probability that an element remains in the observed part of the system is one minus the probability that it has left by time t , the CCDF of the geometric distribution.
- Throughout the paper and supplement the reader is told that the model fits the data or the approximation fits the simulations "well" or "very well". This confuses me because I don't know what the yardstick is.
- Text between S26 and S27 the word "do" can be struck.
- I suggest that the authors consider an alternative for the word "success" as we humans rank a great many things, and for many of those things, being and staying at the top of the list may have the opposite valence. What about "inertia" or "rank preservation" or something that goes for the concept more directly?

REBUTTAL LETTER FOR

Universal dynamics of ranking

G. Iñiguez, C. Pineda, C. Gershenson, A.-L. Barabási

We thank the Editor and the three reviewers for a thorough consideration of our work. We have taken all comments into account and implemented them as changes in the attached revised manuscript and SI, highlighted in blue. In what follows we present a point-by-point response to all reviewers' comments, with our responses also highlighted in blue.

As a high-level summary, we have performed the following changes:

- A thorough exploration of statistical fluctuations in numerical simulations of the model, with a measure of the systematic bias present in the fitting process and a goodness-of-fit test to decide what datasets follow the universal curve (updated Fig. 3; Section S5.2 and Fig. S18).
- A detailed analysis of the effect of subsampling observations of the ranking list on the behavior of the model, the fitting of model to data, the rate of element replacement, and the position of a dataset over the universal curve (new paragraph in Results, updated Fig. 3; Section S5.3 and Figs. S19-S20 in SI).
- A new paragraph in the Discussion summarizing relevant deviations in the data from the predictions of the model, as well as extensions to the model and future research directions to explore these deviations further.
- A change in terminology, from 'openness' to 'turnover', and from 'success' to 'inertia', according to the suggestions of the reviewers.

REVIEWER COMMENTS

Reviewer #1 (Remarks to the Author):

First of all, I want to thank the authors for a paper that I enjoyed reading. It investigates the dynamic of ranking in a large set of empirical data. It proposes an attractive model based on the notion of evolution or revolution. The evolution process is a controlled process of diffusion. The revolution process consists of long ranking jumps. Results of the experimentation show that real-world systems can be characterized by their rank openness and rank flux. Rank openness measures how fast new elements enter the ranking list. Rank flux estimates the probability that an element enters or leaves the ranking list at a given time. No new elements enter the list in closed systems, while they constantly enter and exit in open systems. The authors show that the top ranks are conserved in real-world open systems. Elements ranked at the top and the bottom of the list is stable in closed systems. They also show that displacement and replacement of elements are sufficient to control this process. The paper is well written and well-organized. The empirical study is very rich. The results are very clearly presented. It will indeed generate interesting discussions on centrality concepts and beyond. These are the strong points for publication.

Response:

We thank the reviewer for a positive assessment of our work, especially the fact that the manuscript was enjoyable reading material, with an attractive model and a rich empirical study, which the reviewer considers as strong points for publication in Nature Communications. We fully agree with the remark about potential consequences on centrality concepts, which we have included in the last paragraph of the Discussion alongside the related Ref. [68].

My minor concerns are the following:

1) I am always a little skeptical about universal laws. I understand that the authors want to put under the same umbrella phenomena of diverse origin. I think it will be interesting to comment a bit on the eventual deviations that may have appeared in the experiments.

Response:

We agree with the reviewer, hence we added a paragraph in the Discussion summarizing the deviations in data from the predictions of the model that we find most intriguing, including ideas on extensions of the model and future research directions that may help close the gap between data and model. Notable deviations are a decrease in flux over time for language datasets (SI Fig. S3), variations in the convexity of the curve

of rank out-flux for very open systems (SI Fig. S4), and the slow decay of rank inertia with long times (SI Fig. S7).

2) The authors claim, "Since rankings mediate access to resources (education, jobs, funds) via public and economic policy, they play a role in social mobility; when misused, they contribute to the disenfranchisement of minorities." I did not fully understand this remark. I think that they should give more explanations or remove this sentence.

Response:

Following the reviewer's remark, we have rewritten this sentence, conveying the idea that rankings play a role in the formation of social hierarchies and the potential rise of systematic inequality. As support we use Refs. [5-7], including the recent paper by Kawakatsu et al. suggested by Reviewer 2.

3) Let's consider that there are multiple jumps at various ranges combined with multiple diffusion at multiple scales. How will that affect the model?

Response:

During the development of the project we considered several modifications to the model along these lines, including a non-uniform selection of elements and ranks in the mechanisms of displacement and replacement (tuned by additional parameters), which may indeed lead to variations in the diffusion process and the nature of the Lévy walks we observe. Another possible extension is to consider rank change with various time scales simultaneously, which would explore a spectrum akin to that between discrete and continuous random walks. We expect model extensions such as these may explain some of the deviations we now describe in the Discussion (non-constant flux, convexity of out-flux, slow decay of rank inertia). For this paper, however, we have decided to focus on the simplest (uniform) version of the model, since it can already emulate most of the features of rank dynamics observed, with plenty of insight despite its simplicity. We plan to explore extensions of the model in future work.

4) Not all readers are familiar with the Savitzky–Golay filter, so it is worth noticing that it is a smoothing filter.

Response:

Thank you for noticing this detail, we have modified the text accordingly, in both the main text (caption of Fig. 2) and in the SI (caption of Fig. S9).

Reviewer #2 (Remarks to the Author):

This manuscript argues that a simple model can recreate simple patterns in the dynamics of rankings. It is an excellent piece of research—at the same time accessible and profound. I recommend it for publication in Nature Communication after some minor changes.

Response:

We thank the reviewer for the positive appraisal of our work, and for the recommendation for publication in Nature Communications.

[Openness rate] I think it would be much more intuitive to call this "turnover rate," and unless the authors have a perfect excuse, please change this. Essentially, this is just the colloquial understanding of turnover rate, whereas openness could mean that the ranking rules are open to change or whatever.

Response:

We fully agree that "turnover rate" is a more appropriate term. We have incorporated the corresponding changes all throughout the main text and SI.

[Citations] This paper seems relevant. It is published after the authors submitted the present manuscript, so I don't criticize its omission, but for the reader, it would be helpful if the authors could discuss it:

Mari Kawakatsu, Philip S. Chodrow, Nicole Eikmeier, Daniel B. Larremore, "Emergence of hierarchy in networked endorsement dynamics"

Proceedings of the National Academy of Sciences Apr 2021, 118 (16) e2015188118

Response:

Thank you for the recommendation. Indeed, this paper is very relevant, particularly in connecting the concepts of ranking lists and the formation of social hierarchies. We now refer to it in the first paragraph of the Introduction (Ref. 6).

[When considering open systems in close interaction with their surroundings...] I think in this paragraph, the authors get carried away too far beyond what their results suggest. "[S]lower' elements provide robustness, while 'faster' elements provide adaptivity." ... for this to connect to grander themes of biology and evolution, the ranking in its entirety must have some meaning, like a genetic code or similar. However, in practice, unlike such a code, only the top of a ranking matters. I think the results in this work are great without speculations like this, so the paper would be stronger if the authors deleted the paragraph.

Response:

We broadly agree with the remark of the reviewer, so we have decided to aim for the middle and move this paragraph to the SI (Section S6). We agree that the paragraph is speculative, but it helps in interpreting the (factual) observation that real-world rank dynamics are heterogeneous, by suggesting that such heterogeneity might convey an evolutionary advantage in preserving the functionality of the system against its environment. This interpretation may only be applicable to some types of ranking lists, hence we changed the paragraph in the SI accordingly. In its place in the main text, we added a new paragraph in the Discussion more in line with the results of our analysis, describing the most relevant deviations seen in data from the predictions of the model (non-constant flux, convexity of out-flux, slow decay of rank inertia), alongside extensions of the model and future directions of research that might help us emulate empirical rank dynamics even more accurately.

Reviewer #3 (Remarks to the Author):

Summary

This is a review of "Universal dynamics of ranking" submitted to Nature Communications for consideration. In this paper, the authors consider two simple mechanisms which describe a dynamics over ordered lists. Either (1) an item from one position is moved to an arbitrary position in the list, chosen uniformly at random, with items between the two positions shifted to account for the move, or (2) an item is deleted and replaced with a new item. The first occurs with probability τ and the second with probability ν , in each time step; the moves are not mutually exclusive. Finally, the authors assume that we do not see ALL of the list (N elements) but only the first N_0 . Through a series of asymptotic approximations, the authors arrive at Eqs (5) and (6) which argue for a universal inverse relationship between τ and ν , after each has been rescaled by p (the relative length of the visible list, N_0/N) and o (the average number of new elements observed over the period of observation). When the authors examine 30 real datasets, fit all their parameters, and plot them in rescaled $\log(\tau)$ $\log(\nu)$ space, the points lie around the line. This leads to the conclusion that there exists a universal dynamics for ranking.

Through the notes that follow, I cannot recommend acceptance of the paper, but hope that the comments are nevertheless useful.

Response:

We thank the reviewer for a very accurate summary of the technical details of our work. We hope that with the corrections and additions we have made to the paper, inspired by the referee's remarks, as well as the clarifications offered below, the reviewer will find our manuscript suitable for publication.

Major Comments

After reading the paper and supplement, I'm not sure what it actually means for understanding real world systems or the dynamics of sorted lists. The top-line conclusion is that real systems must have either high replacement and low movement, or high movement and low replacement, or some combination along the $\text{replacement} \times \text{movement} = 1$ line... but after rescaling by the mean flux, the observed and unobserved system sizes. It sounds like the conclusion is that *any* set of ranking observations, generated by any process or model would, after embedding according to this model, fall along the $\nu \times \tau = 1$ line, more or less. So does the more or less matter? Do we learn something from the fact that Nascar Drivers and Hyenas lie far from the

universal line while the languages lie near it? Are these interesting fluctuations or expected fluctuations? The numerous plots of the supplement indeed show that approximations can recapitulate the means of simulations, but are the fluctuations around the means large or meaningful? In short, this manuscript presents a model as a way to gain understanding of the dynamics of rankings, but the analysis and embedding of the model don't actually end up providing that understanding. In this light, it is hard to see how the paper will be of interest to or influence the field.

Response:

We thank the reviewer for exploring the potential implications of our model, particularly the question of whether fluctuations around the universal curve are meaningful. To answer this question, we have performed an extensive analysis of statistical fluctuations in numerical simulations of the model, which we describe in detail in Section S5.2 of the SI (Fig. S18) and is part of the revised Fig. 3 in the main text.

For a dataset with fitted parameters τ and ν , we run multiple simulations of the model, and fit the model to itself, comparing the resulting distributions of parameters (τ_{sim} and ν_{sim}) with the original fit (τ and ν). Relative differences between simulation and original fit parameter values are always less than 50%, but indeed, the fitting process tends to systematically under- or over-estimate parameter values in the model itself, by a dataset-dependent amount typically lower than 50% (see Fig. S18). This suggests that, while parameter estimates for empirical data don't have large deviations from their 'true' values, dataset positions along the universal curve should be interpreted qualitatively.

We use this analysis to perform a goodness-of fit-test, finding the datasets for which their distance to the universal curve is smaller than for most numerical simulations of the model. It turns out most datasets follow the universal curve in this statistically significant way, except for The Guardian readers (recc), Enron emails, Nascar drivers (Busch), and hyenas. Consistently, these are the datasets visually farther away from the universal curve in Fig. 3a, as the reviewer noticed. We believe the statistical rigor added by these calculations strengthens our results and improves the quality of the manuscript, so we are very thankful for the remark that inspired this new analysis.

The conclusion that "Real world ranking lists are drive either by [large jumps], or [small local movements], both alongside relatively low levels of new elements entering the system," is true, with or without the study. The last clause, regarding the rate at which new elements enter, seems entirely dependent on the frequency of sampling. The remaining paragraphs of the discussion read as broadly speculative. In short, the claims of the paper seem really bold but thinly supported.

Response:

Given these pertinent comments, we have extensively explored the effect of subsampling on the behavior of the model and data, which we describe in detail in the related comment by the reviewer below (see also the related new Section S5.3 and Figs. S19-20 in the SI, a new paragraph in Results, and the extended Fig. 3 in the main text). We find that, when normalizing the fitted probability of element replacement (ν) by the real time between subsampled observations, the rate at which elements enter the system is constant, regardless of the frequency of sampling. This is an interesting finding that we now showcase in Fig. 3, alongside the universal curve.

We agree that the 2nd paragraph in the Discussion is somewhat speculative. Therefore, we have decided to move this discussion about the potential evolutionary benefits of heterogeneous ranking dynamics to the SI (Section S6), highlighting its speculative nature. In its place in the Discussion, we have added a new paragraph exploring the systematic deviations seen in data from the predictions of the model, their possible origins, and potential extensions to the model that may increase its accuracy. We believe these additions better support the claims of our manuscript.

More constructively, I suggest that the authors consider a venue that allows their analysis more room to breathe. The paper was not understandable without the supplementary material, but the supplementary material was clearly written and really explicated the model nicely.

Response:

We thank the reviewer for acknowledging that the SI was written well and managed to describe the model nicely. Indeed, our goal was to use the main text to give an overview of the most important results of the analysis, and use the SI to present extensive mathematical derivations and minor considerations, without worry on space restrictions.

Sections S3 and S4 were engaging and interesting. I very much enjoyed reading the mathematics of the paper, yet the density of the writing and notation really got in the way of the communication of ideas, leading me to create notecards to follow along. For instance, what is called a Levy or Levi flight with power law exponent 0 is a complicated way of saying that an element moves to a position in the list chosen uniformly at random.

Response:

We thank the reviewer for a positive assessment of the SI, particularly regarding the mathematical derivations. We agree that the text and notation might feel at times dense. Part of the reason is the large amount of quantities we explored at the same time, which made it difficult to have consistent notation. Still, we made an effort to simplify it as

much as possible, arriving at the present version. We also agree that the reference to Lévy walks might seem like a complicated way to refer to uniform movement. Still, we wanted to highlight the connection between our model and well-known concepts in random walks, a highly explored area in statistical physics. Indeed, if we would extend the model to allow for non-uniform sampling of elements and ranks in the mechanisms of replacement and displacement, the emergent jumps in rank space might be broadly distributed with a tunable exponent, which makes the connection to Lévy walks even more apparent. We have added this discussion to Section S4.2 in the SI.

Another suggestion to improve the analysis would be to consider whether there are available statistical approaches to model fitting, rather than graphical/numerical fits to asymptotic approximations of mean behavior. Given a dataset, and the authors' posed generative process, could one simply find MLE estimates of τ and ν ? If the generative process is available, could one determine whether real systems are within the expected fluctuations of the model or whether the dynamics doesn't apply well?

Response:

We thank the reviewer for this very interesting suggestion. Indeed, it would be ideal to have a more formal statistical approach to model fitting. We therefore attempted to derive MLE equations for two probability distributions: (i) The distribution of the time t an element stays in the ranking list (the product of mean turnover rate in Eq. 4 and rank inertia in Eq. S43); and (ii) the probability of element displacement in Eq. S21. Unfortunately, the MLE equations (for τ and ν) coming from maximizing the log-likelihoods of these two distributions are transcendental and cannot be solved analytically. We solved the MLE equations numerically and compared their solutions to the fitted ν and τ from our 'mean-field' fitting process. For most datasets, mean-field parameter estimates lead to lower mean-squared-error deviations between the fitted model and 6 measures in the data (rank flux, turnover, out-flux, change probability, rank diversity, and inertia). This means that our current fitting process is quite good at recovering some aggregate features of the data, despite it being based on asymptotic approximations of mean behavior. As added value, the mean-field fitting process is more analytically tractable, computationally faster to obtain, and provides insight on the balance between amounts of element displacement and replacement seen in data (i.e. the universal curve). For these reasons, we have decided to focus on our original fitting method for the time being. In future work, we can attempt to use more sophisticated statistical approaches, such as ABC methods or Bayesian optimization that do not require likelihood functions for inference. We have added this discussion to the SI (end of Section S5.2).

We agree completely with the reviewer in that, regardless of whether MLE is used or not, a measure of expected fluctuations and statistical significance in the model was

required. As described in a previous comment, we have calculated the expected variation in parameter estimates for the model itself, and used it to perform a goodness-of-fit test, identifying the systems for which the model applies well, in the sense that the dataset is closer to the universal curve than most simulations of the fitted model (see Section S5.2 of the SI and Fig. S18). We wish to thank the reviewer for initiating this work.

Finally, the authors have the ability to evaluate how real data points move along the "universal" curve in the presence of subsampling (say, discarding every other or every third sample of ranks) or arbitrarily shifting upward the line at which rank flux is calculated. Showing how the inferred τ and ν shift, for a single underlying system, when viewed through different censoring or scoping, would help future readers understand how sampling rate and sample scope of the data affects the inferences one draws from it.

Response:

We thank the reviewer for this very interesting suggestion. We have performed an extensive analysis on the effect of subsampling in the behaviour of model and data (see the related Section S5.3 and Figs. S19-20 in the SI, a new paragraph in Results, and the extended Fig. 3 in the main text). We implement subsampling in the way suggested by the reviewer, only considering every k -th observation of the ranking in model or data, with k a tunable parameter. In the model, we derive approximate expressions for the effective parameters (τ_k and ν_k) driving the dynamics after subsampling, and find that they increase almost linearly with k (from their original values τ and ν for $k=1$). Rank flux and turnover also increase with subsampling, while rank inertia decreases. Similar effects appear when subsampling empirical data, which we use to show how datasets move downwards along the universal curve, as the reviewer suspected. Apart from the fluctuation analysis, this is another way of qualitatively assessing the goodness of fit of the model: most datasets follow the universal curve even in the presence of subsampling.

This analysis also uncovered an interesting aspect of subsampling. The rate of element replacement (i.e. the fitted replacement probability ν normalized by the real time between subsampled observations) is constant with respect to k (see new Fig. 3b in the main text, and Table S2 in the SI). Moreover, this quantity 'clusters' systems of different types, showing that online social systems (GitHub, The Guardian) replace elements the most often, followed by sports, with languages the most stable systems in terms of rates of element replacement, regardless of subsampling. Indeed, this suggestion by the reviewer further uncovered consequences of our analysis that shed light on generic features of rank dynamics, which have increased the value of the study.

Minor comments

- As mentioned above, I suggest the authors consider not invoking Levy flights. If they feel it is necessary to do so, I suggest consistently writing Levi or Levy, but not both.

Response:

Thank you for noticing this detail. We have decided to keep the term Lévy flight, but we have now corrected the text and consistently used the word 'Lévy' throughout the main text and SI.

- S4.1 the idea of "left/right" is introduced before it is explained in S4.2. A simple reordering could fix.

Response:

We have implemented this change.

- Though identical, perhaps an simpler alternative to deriving S8 is to go directly at it: the probability that an element remains in the observed part of the system is one minus the probability that it has left by time t , the CCDF of the geometric distribution.

Response:

We thank the reviewer for noticing this detail. We have changed the corresponding text accordingly.

- Throughout the paper and supplement the reader is told that the model fits the data or the approximation fits the simulations "well" or "very well". This confuses me because I don't know what the yardstick is.

Response:

We agree with the reviewer that these expressions are somewhat ambiguous. What we imply in general is qualitative agreement between the functional shape of a quantity and its approximation in the model. We have changed the text accordingly all across the manuscript. We also hope that the added analysis (fluctuations in the model, goodness-of-fit test, and effect of subsampling) gives a more accurate picture of the quantitative comparison between model simulations, model approximations, and empirical data.

- Text between S26 and S27 the word "do" can be struck.

Response:

Thank you for noticing this detail. We have fixed it.

- I suggest that the authors consider an alternative for the word "success" as we humans rank a great many things, and for many of those things, being and staying at the top of the list may have the opposite valence. What about "inertia" or "rank preservation" or something that goes for the concept more directly?

Response:

We agree. We now use the term 'rank inertia', or 'inertia' for short, throughout the SI.

REVIEWER COMMENTS

Reviewer #1 (Remarks to the Author):

All the remarks have been well addressed. Congratulations.

Reviewer #2 (Remarks to the Author):

The authors have adequately addressed my concerns, so I recommend the manuscript for publication.

Reviewer #3 (Remarks to the Author):

Summary

This is a review of "Universal dynamics of ranking" resubmitted to Nature Communications for consideration. The authors have made considerable changes to the manuscript, and have responded to almost all of the concerns that I and other reviewers have raised. However, as I explain below, I am concerned that the conclusions of this modeling effort are not supported by the evidence the authors present. I cannot recommend publication, in spite of the enthusiasm of the other reviewers.

Issues

A. The following issue is technical but critical. I explain in full detail, and apologize for any over-explanation. In short, the values plotted in Fig 3a, on which the paper's core claim of universality rests, are not reliably inferred from the data. In other words, the paper's claims rest on fitting a model to the data, but because that process seems to be critically flawed, it is hard to see where that leaves the paper's primary conclusion. In detail:

The fitting of the model to data suffers from uncontrolled and unexplored bias, calling Fig 3a into question. The new Figure S18 actually illustrates the issue clearly. The authors have performed what statisticians would call a parametric bootstrap:

- 1. From a sample (data), fit the parameters of the model (here: τ , ν).**
- 2. To estimate uncertainty around those fitted parameters, simulate multiple synthetic datasets using the fitted parameters, creating new data with variation that comes from the model's stochasticity.**
- 3. Create a bootstrapped distribution of fitted parameters by fitting the model to each of the synthetic datasets. (here: τ_{sim} , ν_{sim})**

In principle, the τ_{sim} and ν_{sim} distributions help us understand how much variation to expect around the originally estimated τ and ν , leading to, for instance, a bootstrapped confidence interval. The authors show these bootstrapped distributions in kernel density estimates in Fig S18, after subtracting off the original estimates and rescaling (as noted in the caption).

If one estimates a parameter and then bootstraps (parametrically or nonparametrically) a distribution of estimates, it is problematic if the distribution doesn't contain the original estimate. For comparison, consider estimating a mean and then bootstrapping a CI for the

mean, only to find that the CI doesn't contain the average of the data! One can quantify this issue by asking how often a bootstrapped parameter is smaller than the original data-fit parameter. One might expect a value around 0.5, because, by chance, some of the synthetic data will lead to smaller parameters, and other data to larger ones. If the distribution is skewed, perhaps the value won't be exactly 0.5.

The authors have actually calculated a version of this value, and call it Lambda (Fig S18), but unfortunately it undermines their broader analysis and conclusions. In effect, it shows exactly how often the bootstrapped distributions seem to plausibly contain their own generating parameters: only 1 out of 30 datasets (Nascar 0.42 ± 0.01). This tells us that the process of taking parameters, generating data, and fitting parameters is actually biased. The loop doesn't close.

The authors transparently note this bias. But what is not discussed are the conclusions that must be drawn from S18 and its Lambdas: one cannot reliably generate data from the proposed theoretical model of dynamics and then infer said parameters from data. In other words, we cannot actually trust our ability to fit the model to the data, even when we generate the data using the true model. This is a bias in model estimation, and it tells us that the model does not behave the way unbiased estimators are supposed to. It is akin to drawing samples from a standard normal distribution, using a method to compute their average, and finding that said average is greater than zero every time.

If we cannot reliably fit the model to the data without bias, even in ideal synthetic conditions, it is impossible to argue that we should interpret the values plotted in Fig 3a, which underpin the claims of a universal dynamics of ranking in real data.

(One alternative possibility is that there is a bug in the code used to generate synthetic data, which causes the generating parameters and the inferred parameters not to align.)

B. In addition to (A), the goodness of fit test used in the revision is, itself, not a reasonable test. The revision quantifies the number of bootstrapped parameters that move toward the theory line, and concludes that 87% (26/30) of datasets fit the data. Setting aside the problem that it seems hard to claim universality when 87% (binomial 95% CI: [69%, 96%]) of datasets exhibit the hypothesized universal behavior, a test of whether the data plausibly follow the theory would more directly ask whether the bootstrapped distribution includes the hypothesized point at $\tau_r * \nu_r = 1$. If the bootstrapped distribution doesn't include the theoretical point, one would be inclined to conclude that the data deviates from the theory. This analysis is not what the authors have done, so we remain unable to evaluate the claims. The methodological issues in (B) are orthogonal to the issues in (A).

C. In addition to the issues above, I again object to the fact that uniform random movement is characterized as a Levy flight. I think this is likely to confuse readers. It is ok to call a uniform distribution a uniform distribution.

D. All other issues raised in the first reviews seem to be addressed. I thank the authors for their thorough exploration (particularly of the data subsampling question).

Unfortunately, based primarily on A and B, it is impossible to conclude that the authors have indeed identified a universal dynamics of rankings. The manuscript presents an interesting idea, but one which does not appear to be borne out in the analyses.

REBUTTAL LETTER FOR

Universal dynamics of ranking

G. Iñiguez, C. Pineda, C. Gershenson, A.-L. Barabási

We thank the Editor and the three reviewers for a thorough consideration of our work. We have taken all comments into account and implemented them as changes in the attached revised manuscript and SI, highlighted in blue. In what follows we present a point-by-point response to all reviewers' comments, with our responses highlighted in blue.

As a high-level summary, we have performed the following changes:

- A rewritten Section S5.2 in the SI, to take into account the concerns of reviewer #3 related to bias in the estimation of model parameters, alongside new Figs. S18, S19, and S20.
- A corrected parametric bootstrap process (together with a more detailed description of it) that leads to the unbiased estimation of parameters p and ν , and a small bias in parameter τ . Evidence that, despite this bias, numerical simulations of the model correctly capture aggregate features of empirical data (rank flux, turnover, and inertia).
- Minor related changes in the revised manuscript.

REVIEWER COMMENTS

Reviewer #1 (Remarks to the Author):

All the remarks have been well addressed. Congratulations.

Reviewer #2 (Remarks to the Author):

The authors have adequately addressed my concerns, so I recommend the manuscript for publication.

We thank reviewers #1 and #2 for a positive appraisal of our revised manuscript, and for recommending its publication in Nature Communications.

Reviewer #3 (Remarks to the Author):

Summary

This is a review of "Universal dynamics of ranking" resubmitted to Nature Communications for consideration. The authors have made considerable changes to the manuscript, and have responded to almost all of the concerns that I and other reviewers have raised. However, as I explain below, I am concerned that the conclusions of this modeling effort are not supported by the evidence the authors present. I cannot recommend publication, in spite of the enthusiasm of the other reviewers.

We thank the reviewer for a continuing and careful consideration of our manuscript. We hope that the changes included in this revised version (and described in detail below) will fully address these concerns.

Issues

A. The following issue is technical but critical. I explain in full detail, and apologize for any over-explanation. In short, the values plotted in Fig 3a, on which the paper's core claim of universality rests, are not reliably inferred from the data. In other words, the paper's claims rest on fitting a model to the data, but because that process seems to be critically flawed, it is hard to see where that leaves the paper's primary conclusion. In detail:

The fitting of the model to data suffers from uncontrolled and unexplored bias, calling Fig 3a into question. The new Figure S18 actually illustrates the issue clearly. The authors have performed what statisticians would call a parametric bootstrap:

1. From a sample (data), fit the parameters of the model (here: τ , ν).
2. To estimate uncertainty around those fitted parameters, simulate multiple synthetic datasets using the fitted parameters, creating new data with variation that comes from the model's stochasticity.
3. Create a bootstrapped distribution of fitted parameters by fitting the model to each of the synthetic datasets. (here: τ_{sim} , ν_{sim})

In principle, the τ_{sim} and ν_{sim} distributions help us understand how much variation to expect around the originally estimated τ and ν , leading to, for instance, a bootstrapped confidence interval. The authors show these bootstrapped distributions in kernel density estimates in Fig S18, after subtracting off the original estimates and rescaling (as noted in the caption).

If one estimates a parameter and then bootstraps (parametrically or nonparametrically) a distribution of estimates, it is problematic if the distribution doesn't contain the original estimate. For comparison, consider estimating a mean and then bootstrapping a CI for the mean, only to find that the CI doesn't contain the average of the data! One can quantify this issue by asking how often a bootstrapped parameter is smaller than the original data-fit parameter. One might expect a value around 0.5, because, by chance, some of the synthetic data will lead to smaller parameters, and other data to larger ones. If the distribution is skewed, perhaps the value won't be exactly 0.5.

The authors have actually calculated a version of this value, and call it Lambda (Fig S18), but unfortunately it undermines their broader analysis and conclusions. In effect, it shows exactly how often the bootstrapped distributions seem to plausibly contain their own generating parameters: only 1 out of 30 datasets (Nascar 0.42 ± 0.01). This tells us that the process of taking parameters, generating data, and fitting parameters is actually biased. The loop doesn't close.

The authors transparently note this bias. But what is not discussed are the conclusions that must be drawn from S18 and its Lambdas: one cannot reliably generate data from the proposed theoretical model of dynamics and then infer said parameters from data. In other words, we cannot actually trust our ability to fit the model to the data, even when we generate the data using the true model. This is a bias in model estimation, and it tells us that the model does not behave the way unbiased estimators are supposed to. It is akin to drawing samples from a standard normal distribution, using a method to compute their average, and finding that said average is greater than zero every time.

If we cannot reliably fit the model to the data without bias, even in ideal synthetic conditions, it is impossible to argue that we should interpret the values plotted in Fig 3a, which underpin the claims of a universal dynamics of ranking in real data.

(One alternative possibility is that there is a bug in the code used to generate synthetic data, which causes the generating parameters and the inferred parameters not to align.)

We appreciate the detailed explanation, which has actually helped us understand more in depth the nature of our fitting process. We point the referee to the revised Section S5.2 (and related Figs. S18, S19, and S20), which we have modified to take all these considerations into account.

As the referee points out, we have performed a parametric bootstrap of the model: For a dataset with fitted model parameters τ and ν , we run model simulations with the

same parameters, fit the model to itself, and obtain distributions of the bootstrapped parameter τ_{sim} and ν_{sim} .

We have made a couple of corrections to the bootstrap process:

- (1) In order to run simulations appropriate for each dataset, we need not only the same τ and ν , but the same number of observations T , list size N_0 , and total number of elements ever seen in ranking N_{T-1} (thus fixing the fraction p as in the data). T and N_0 can be fixed in advance, but N_{T-1} can only be determined after running the simulation for some undetermined system size N . In the previous version of the bootstrap we used the approximation for turnover (Eq. S38) to make a guess for N (Eq. S52 in the previous version of the SI), which introduced a systematic error in the simulated N_{T-1} . In the new bootstrap process, we run simulations with variable N and choose the value that on average minimizes the difference in N_{T-1} between data and simulations. With N fixed, we run simulations again and fit the model to itself to obtain τ_{sim} and ν_{sim} .
- (2) In the previous version of Fig. S18 we plotted the bootstrapped parameter distributions in terms of the relative parameter difference $(\tau_{\text{sim}} - \tau) / \tau$ (and similar for ν), a rescaling that amplifies differences particularly when τ or ν are small. Since τ and ν are already probabilities (i.e. bounded in $[0,1]$), this rescaling is not necessary and now we use the raw difference $\tau_{\text{sim}} - \tau$ (and similar for ν).

Results are shown in the new Fig. S18 (both as binned histograms and KDEs). By construction, the bootstrapped distribution of p_{sim} contains p (step 1 described above). Similarly, the distribution of ν_{sim} contains ν (in all datasets except Hyenas), meaning that the estimation of ν is actually unbiased. Even for Hyenas, the difference between bootstrapped mean and original estimate in ν is very small. The estimation of τ is, on the other hand, biased: many datasets show a small nonzero difference between bootstrapped mean and original estimate (never larger than 0.1), and some of these distributions don't include the original τ .

Here we would like to point out that our goal is not to build an unbiased estimator of the parameters of the model, or for that matter, to devise a statistical test that fails to reject the null hypothesis that our model generates the empirical data. Given the simplicity of the model, it's likely we'll eventually find a test statistic that shows a significant deviation between model and data. Our goal is to show that, despite a small bias in parameter estimation, the fitted model can actually recover aggregate features of ranking dynamics (flux, turnover, and inertia), suggesting an underlying universality in the mechanisms driving real-world ranking dynamics.

We have changed the rationale of Section S5.2 to reflect this viewpoint. Fig. S19 now shows the bootstrapped differences in flux, turnover and inertia between model simulations and empirical datasets. Despite the bias in estimating tau, simulations with fitted model parameters recover turnover perfectly, and recover flux up to a difference of less than 0.1 between bootstrapped mean and empirical value. To a lesser degree, simulations even recover quantities not explicitly involved in the fitting process, such as inertia (up to a difference of 0.1, like in the case of flux).

We would also like to point out a misunderstanding in the interpretation of the quantity Lambda, probably due to how briefly we discussed it in the SI. Lambda is not a measure of whether or not the bootstrapped distributions contain their own generating parameters; this is better captured by the distributions in the new Fig. S18 crossing the vertical line at 0, which happens in all datasets for p, in all but Hyenas for nu, and only in a few datasets for tau (due to bias).

In order to clarify the text, we have removed the explicit mention of Lambda and now include Fig. S20. Here we concentrate on the rescaled fitted parameters tau_r and nu_r (single values for each dataset, or bootstrapped distributions for model simulations) and compute their relative difference to the ideal behavior ($\tau_r * \nu_r = 1$) in either tau or nu axis, i.e. $\tau_r * \nu_r - 1$ (see new Eq. S52). We see that for most datasets, the distance to the universal curve for data (i.e. for the dots in Fig. 3a) is actually smaller than for simulations of the model fitted to itself, suggesting that empirical data follows the universal curve. This was the intended use of Lambda; to count the number of simulations farther away from the vertical line at 0 than the data line.

B. In addition to (A), the goodness of fit test used in the revision is, itself, not a reasonable test. The revision quantifies the number of bootstrapped parameters that move toward the theory line, and concludes that 87% (26/30) of datasets fit the data. Setting aside the problem that it seems hard to claim universality when 87% (binomial 95% CI: [69%, 96%]) of datasets exhibit the hypothesized universal behavior, a test of whether the data plausibly follow the theory would more directly ask whether the bootstrapped distribution includes the hypothesized point at $\tau_r * \nu_r = 1$. If the bootstrapped distribution doesn't include the theoretical point, one would be inclined to conclude that the data deviates from the theory. This analysis is not what the authors have done, so we remain unable to evaluate the claims. The methodological issues in (B) are orthogonal to the issues in (A).

From our point of view, it is actually remarkable that 26/30 datasets show common features of rank dynamics that also arise in a very simple generative model, especially

when we didn't engage in any pre- or post-selection criteria apart from having enough data to define the rank of elements at snapshots in time (see beginning of Results section in the manuscript and Section S2 in the SI). In line with our response above, we agree with the referee in not using the Lambda analysis as a goodness of fit test. Rather, we now frame Section S5.2 as detailed evidence that the fitted model recovers flux, turnover and inertia in all datasets, up to small errors and despite a bias in estimating tau.

We believe that an appropriate test would ask 'how close' is the bootstrapped distribution to the universal curve ($\tau_r * \nu_r = 1$), rather than if the distribution 'contains' the universal curve. This because the universal curve (Eq. 6, Eq. S48) is in itself an approximation of the actual behavior of the model (see the right side of Fig. S16, where the continuous line drops down away from the dashed line with slope -1). The expressions for flux and turnover constituting the fitting process (Eqs. S27 and S40) are also approximations. Thus, if the bootstrapped distribution doesn't contain the universal curve, it's difficult to distinguish whether this happens because the approximations fail, or because the model is incapable of recovering ranking dynamics in data.

With this in mind, the new Fig. S20 explores the relative difference ($\tau_r * \nu_r - 1$) between the universal curve and either simulations or empirical data. The bootstrapped distribution shows a relative difference of up to 0.5 in magnitude, which is partly due to the bias in tau. Still, the fitting process is such that data is actually closer to the universal curve than simulations for most datasets. We present this as evidence that empirical datasets follow the universal curve, up to variations in τ_r (y-axis in Fig. 3a) due to the estimation bias in tau.

Overall, we would like to stress the fact that our claim to universality doesn't only come from whether datasets follow the universal curve or not, but from the fact that even a model this simple can replicate the functional shape and parameter dependency of several measures of empirical rank dynamics (rank flux, turnover, change, inertia, and the displacement probability), as expressed throughout the main text and SI.

C. In addition to the issues above, I again object to the fact that uniform random movement is characterized as a Levy flight. I think this is likely to confuse readers. It is ok to call a uniform distribution a uniform distribution.

We have changed the text after Eq. 1 to take into account the comment of the referee (also note the highlighted text in the SI after Eq. S5). We think, however, that there is some gain in using the concept of Levi walk, with a proper explanation of what we

mean, to avoid confusing the audience. This is particularly apparent when using Levi walk as a shorthand for the low- ν_r , high τ_r regime in Fig. 3a (left upper part of the universal curve). This regime is characterized by potentially large changes in rank (corresponding to the intuition of a large step length in a Levi walk), as opposed to the small changes in rank (due to the movement of other elements) found in the diffusion regime, further down along the universal curve (see also the pink/green areas in Fig. 2b). The use of the Levi walk shorthand highlights the contrast between regimes of long vs. short rank jumps, which is one of the features of empirical ranking lists uncovered by the model.

D. All other issues raised in the first reviews seem to be addressed. I thank the authors for their thorough exploration (particularly of the data subsampling question).

We thank again the referee for proposing the subsampling analysis, which led us to uncover yet another generic feature of rank stability: a rate of element replacement robust to changes in the sampling rate.

Unfortunately, based primarily on A and B, it is impossible to conclude that the authors have indeed identified a universal dynamics of rankings. The manuscript presents an interesting idea, but one which does not appear to be borne out in the analyses.

We hope that this revised version, alongside our responses above, will clarify the results we present to suggest a universal dynamics of empirical ranking lists.